# A Score-Based Approach for Training Schrödinger Bridges for Data Modelling

**DOI:** 10.3390/e25020316

**Published:** 2023-02-08

**Authors:** Ludwig Winkler, Cesar Ojeda, Manfred Opper

**Affiliations:** 1Machine Learning Group, Technische Universität Berlin, 10623 Berlin, Germany; 2Artificial Intelligence Group, Technische Universität Berlin, 10623 Berlin, Germany; 3Institute of Mathematics, University of Potsdam, 14469 Potsdam, Germany; 4Centre for Systems Modelling and Quantitative Biomedicine, University of Birmingham, Birmingham B15 2TT, UK

**Keywords:** Schrödinger bridge problem, score estimation, reverse-time stochastic processes

## Abstract

A Schrödinger bridge is a stochastic process connecting two given probability distributions over time. It has been recently applied as an approach for generative data modelling. The computational training of such bridges requires the repeated estimation of the drift function for a time-reversed stochastic process using samples generated by the corresponding forward process. We introduce a modified score- function-based method for computing such reverse drifts, which can be efficiently implemented by a feed-forward neural network. We applied our approach to artificial datasets with increasing complexity. Finally, we evaluated its performance on genetic data, where Schrödinger bridges can be used to model the time evolution of single-cell RNA measurements.

## 1. Introduction

Recently, there has been an increasing interest in the application of continuous-time stochastic processes as generative data models, e.g., the so-called *diffusion models* have recently achieved state-of-the-art performance in generating data with unmatched fidelity and granularity [1,2,3,4]. This approach to generative modelling proceeds by incrementally adding noise to the original data until they are indistinguishable from samples drawn from a prior distribution, which one selects such that it is easy to sample from, e.g., the Gaussian distribution. One then is required to learn the denoising procedure, whereby one is able to recover from the prior the original data distributions. The authors in [1] showed that one can model the noising procedure as a stochastic process in terms of a stochastic differential equation (SDE), whose drift function is defined such that the desired prior becomes the marginal distribution of the process at the end time. If one defines the noise-injecting process as the forward process, an equivalent SDE, which runs backward in time from the prior to the data, can be used as the denoiser to generate data samples. This approach can be made computationally tractable because the drift of the backward process at each time is given explicitly in terms of the score function, which equals the gradient of the logarithm of the marginal density of the forward process. Hence, if in the forward process, one uses simple Gaussian transition probabilities (related to Ornstein–Uhlenbeck processes), the required scores can be estimated from exact samples and be represented by neural networks as function approximators.

A more general version of generative models is based on the so-called Schrödinger bridges. Here, one tries to construct a stochastic process that has *two* given distributions as marginals at the initial and end times. In addition, the process has to stay close (in a probabilistic sense) to a reference process. While this scenario can obviously be applied to a Gaussian prior and a data distribution, it allows for more general applications, where, e.g., one tries to interpolate between two different arbitrary data distributions.

Schrödinger bridges were introduced by E. Schrödinger in 1931 [5,6] as the most-likely temporal evolution of the probability density for diffusing particles between given initial and end distributions. Recent reviews of the theoretical foundations, formulations as a stochastic control problem, and relations to optimal transport theory can be found, e.g., in [7,8]. Connections to particle-based filtering and smoothing problems occurring in the field of data assimilation were reviewed in [9].

In contrast to the previously mentioned diffusion models, the solution of Schrödinger bridge problems is computationally more demanding. Feasible methods are usually based on the so-called Iterative Proportional Fitting (IPF) algorithm [10] (also known as the Sinkhorn algorithm), which can be understood from the formulation of the bridges as entropically regularised optimal transport problems.

IPF solves the bridge problem by creating a convergent sequence of simpler forward and backward processes, known as half-bridges. For those sub-problems, only *one* of the two distributions at the boundaries of the time interval is kept fixed (alternating between initial and end-points). Recent algorithms differ in the way these half-bridge problems are solved. The authors of [11] presented a method that is based on sequential Monte Carlo techniques for efficiently sampling processes in both directions. References [12,13] are, to our knowledge, the first papers to discuss Schrödinger bridges as generative data models from a machine learning perspective. The construction of the half-bridges of the IPF algorithm is formulated in terms of an estimation for the drift functions of the corresponding SDE. A drift functions is learned from samples created by the half-bridge of the previous iteration using either a Gaussian process regression approach or by training a deep neural network.

The estimation of the required (backward) drift functions from samples of the forward processes (and vice versa) is less simple compared to the case of diffusion models. The exact representation of reverse drifts in terms of score functions could in principle be applied to the Schrödinger problem by a sample-based estimation of score functions using the so-called *score matching* method [14]. However, this approach was deemed as computationally too demanding [12] because it would represent the drift at a given iteration as a sum of scores obtained in the previous steps. Using neural networks as function approximators for score functions would require the storage of an increasing number of neural networks, two per iteration. References [12,13], in contrast, used methodologies that explicitly rely on the *Euler–Maruyama* (EM) discretisation of the SDE. Reference [12] derived an approximation for score functions that allows for an estimation of the reverse drift as a regression problem involving the states at neighbouring discrete times. In a similar way, the approach [13] uses likelihood approximation based on EM for drift estimation. In principle, both estimators depend on the temporal discretisation, which adds another approximation to the solution of the Schrödinger bridge problem.

In this paper, we developed a novel score-based approach to solving the half-bridge problems for Schrödinger bridges within the IPF algorithm. It relies on the exact representation of the drift for the backward process in terms of forward drift and a score function. We developed a variational formulation, which generalises the original score matching approach and which does not rely on the EM discretisation. It is based on a cost functional with a minimiser that agrees with the reverse drift. A sample-based empirical approximation of the cost functional can be minimised using a neural-network-based representation of the drift function. This requires the storage of only a single neural network during the algorithm. We evaluated the quality of our method on two synthetic datasets, as well as on a single-cell RNA sequencing benchmark problem and, for the latter case, showing an improvement upon previous methods by a significant margin.

## 2. Materials and Methods

In the following subsections, we review the definition of the Schrödinger bridge problem. We discuss how it can be solved by the *Iterative Proportional Fitting* (IPF) method, which relies on estimating drift functions for time-reversed stochastic processes. We show how this estimation can be performed using a modified score matching estimator. Finally, we discuss the practical implementation of the estimator using a feed-forward neural network.

### 2.1. Stochastic Differential Equations

We considered the dynamics of a *D*-dimensional state variable $\left\{X_{t}:0\leq t\leq 1\right\}$ in continuous time *t*, which is defined by a stochastic differential equation (SDE) [15] of the form: (1)$$\begin{array}{c} dX_{t}=\vec{\mu }\left(X_{t},t\right)dt+\sigma dW_{t} \end{array}$$
Here, $dX_{t}$ is the change of $X_{t}$ during an infinitesimal time interval dt. $\vec{\mu }(X_{t},t)$ is the drift function, i.e., the deterministic part of the driving force. We introduce the harpoons to indicate the direction of integration, as we will later on introduce reverse-time stochastic processes. $\sigma dW_{t}$ denotes the diffusion part, which describes a stochastic, white noise force term defined by the infinitesimal change of a Wiener process $W_{t}$, where the diffusion strength is set by the constant scalar σ. The formal definition of the drift is given by the (conditional) expected infinitesimal change of $X_{t}$ by
(2)$$\begin{array}{c} \vec{\mu }\left(x,t\right)=\lim_{h\to 0}\frac{1}{h}E\left[X_{t+h}-X_{t}|X_{t}=x\right] \end{array}$$

### 2.2. Schrödinger Bridge Problem

We here give a short, informal introduction to the Schrödinger bridge problem. A more detailed and rigorous discussion can, e.g., be found in [8]. The Schrödinger bridge problem consists of constructing an SDE (with fixed given diffusion σ) such that the probability densities of the corresponding state variables $X_{t=0}$ and $X_{t=1}$ at initial and final times (which, for simplicity, we take to be t=0 and t=1) coincide with given densities $\pi_{0}\left(x\right)$ and $\pi_{1}\left(x\right)$. In order to make the problem unique, one imposes the additional constraint that the probability measure P over the corresponding paths of the stochastic process should be close to a given reference measure $Q_{0}$. The latter is itself defined by a drift function ${\vec{\mu }}^{Q0}\left(X_{t},t\right)$ and (for simplicity) a given *initial* density $\pi_{0}\left(x\right)$.

If we define $D(\pi_{0},\pi_{1})$ to be the set of probability measures over paths $\left\{X_{t}:0\leq t\leq 1\right\}$ with fixed marginal densities $\pi_{0},\pi_{1}$, the measure $P^{*}$ over paths of the SDE that solves the Schrödinger bridge is defined by the solution of the minimisation problem: (3)$$\begin{array}{c} P^{*}=\arg\underset{P\in D(\pi_{0},\pi_{1})}{\inf}\mathrm{KL}\left[\;P\;||\;Q_{0}\;\right]. \end{array}$$
The explicit expression of the KL-divergence between two different path measures P and Q (with the same σ) induced by two SDEs with drift functions ${\vec{\mu }}^{P}\left(x,t\right)$ and ${\vec{\mu }}^{Q}\left(x,t\right)$ and *initial* densities $\pi_{0}^{P}\left(x\right)$ and $\pi_{0}^{Q}\left(x\right)$ (for $X_{t=0}$) is given by
(4)$$\begin{array}{c} \mathrm{KL}\left[\;P\;||\;Q\;\right]=\mathrm{KL}\left[\;\pi_{0}^{P}\;\left|\right|\;\pi_{0}^{Q}\;\right]+\frac{1}{2\sigma^{2}}\int_{0}^{1}E_{P}\left[{\left({\vec{\mu }}^{P}\left(X_{t},t\right)-{\vec{\mu }}^{Q}\left(X_{t},t\right)\right)}^{2}\right]dt \end{array}$$
where
(5)$$\begin{array}{c} \mathrm{KL}\left[\;\pi_{0}^{P}\;\left|\right|\;\pi_{0}^{Q}\;\right]=\int \pi_{0}^{P}\left(x\right)\ln\left(\frac{\pi_{0}^{P}\left(x\right)}{\pi_{0}^{Q}\left(x\right)}\right)dx \end{array}$$
denotes the usual KL-divergence between probability densities in $R^{D}$. Hence, the Schrödinger bridge problem can be understood as a problem of optimal stochastic control, where one has to find a drift function ${\vec{\mu }}^{P^{*}}\left(x,t\right)$ as a state and time-dependent control variable that steers the stochastic dynamical system (Equation 1) in such a way that the marginal density of the state variable evolving from a given initial density reaches a predefined end density. In addition, control variables are quadratically penalised by the KL-divergence (Equation 4) to stay close on average to the drift of the reference system ${\vec{\mu }}^{Q}\left(x,t\right)$.

### 2.3. Iterative Proportional Fitting in Schrödinger Bridges

A popular methodology to solve the Schrödinger bridge problem is via Iterative Proportional Fitting (IPF) [16,17], which solves the problem iteratively, where in each iteration step *i*, two so-called *half-bridge* problems have to be solved. These half-bridges are defined by the recurrent optimisation problems: (6)$$\begin{array}{cc} P_{i}^{*} & =\arg\underset{P\in D(\cdot ,\pi_{1})}{\inf}\mathrm{KL}\left[\;P\;||\;Q_{i-1}^{*}\;\right] \end{array}$$(7)$$\begin{array}{cc} Q_{i}^{*} & =\arg\underset{Q\in D(\pi_{0},\cdot )}{\inf}\mathrm{KL}\left[\;Q\;||\;P_{i}^{*}\;\right] \end{array}$$
for i=1,2,… with an initial measure defined by the reference process, i.e., $Q_{0}^{*}=Q_{0}$ for i=1. In the first half-bridge, one minimises the KL-divergence with only the end condition $\pi_{1}$ fixed, whereas for the second half-bridge, only the initial condition $\pi_{0}$ is fixed. As i→∞, the sequences $P_{i}^{*}$ and $Q_{i}^{*}$ converge to the solution of the Schrödinger bridge problem. For a proof, see [18]. In Figure 1, we provide a visual intuition of IPF applied to the Schrödinger bridge problem.

To solve a half-bridge problem, one can use the fact that a given SDE with drift function $\vec{\mu }(x,t)$ can also be solved backwards in time, where the resulting backward process is also represented by an SDE. We define the reversed time as τ≐1−t and the backward SDE as
(8)$$\begin{array}{c} dZ_{\tau }=\overset{↼}{\mu }\left(Z_{\tau },\tau \right)d\tau +\sigma dW_{\tau } \end{array}$$
with a *backward drift* function that is given by the conditional expectation: (9)$$\begin{array}{c} \overset{↼}{\mu }\left(z,1-t\right)=\lim_{h\to 0}\frac{1}{h}E\left[X_{t-h}-X_{t}|X_{t}=z\right] \end{array}$$
in terms of the forward process $X_{t}$. It can be shown that the statistics of the ensemble of paths $\left\{Z_{1-t}:0\leq t\leq 1\right\}$ coincides with that of the forward process $\left\{X_{t}:0\leq t\leq 1\right\}$ when the initialisation $Z_{\tau =0}$ is drawn at random from the density state variable $X_{t=1}$. The KL-divergence between path measures can also be expressed in terms of the backward processes and drifts as
(10)$$\begin{array}{c} \mathrm{KL}\left[\;P\;||\;Q\;\right]=\mathrm{KL}\left[\;\pi_{1}^{P}\;\left|\right|\;\pi_{1}^{Q}\;\right]+\frac{1}{2\sigma^{2}}\int_{0}^{1}E_{P}\left[{\left({\overset{↼}{\mu }}^{P}\left(Z_{\tau },\tau \right)-{\overset{↼}{\mu }}^{Q}\left(Z_{\tau },\tau \right)\right)}^{2}\right]d\tau \end{array}$$

Equations (Equation 4) and (Equation 10) show that, for given initial or final densities, respectively, the KL-divergences are minimised by matching the drift functions of the processes (the KL-divergences between initial/end marginal densities equal zero). Hence, if we assume that the mapping: (11)$$\begin{array}{c} \vec{\mu }(\cdot ,t)↔\overset{↼}{\mu }(\cdot ,\tau ) \end{array}$$
is known explicitly, the solution of the half-bridges becomes simple. The minimiser $P_{i}^{*}$ of the KL-divergence in Equation (Equation 6) corresponds to an SDE that has the backward drift corresponding to the forward SDE given by the process $Q_{i-1}^{*}$, but is started with the density $Z_{\tau =0}∼\pi_{1}$ in backward time. The same construction holds for $Q_{i}^{*}$ in Equation (7). This is given by a new SDE with a forward drift, which corresponds to the backward drift of $P_{i}^{*}$ and is started from $\pi_{0}\left(x\right)$ in forward time. Hence, the IPF algorithm reduces the Schrödinger bridge problem to the computation of backward and forward drift functions from the corresponding forward and backward processes.

The *explicit relation* between forward and backward drifts was published first [19] and discussed in [20,21] and is given by
(12)$$\begin{array}{cc} \overset{↼}{\mu }(x,\tau ) & =-\vec{\mu }\left(x,1-\tau \right)+\sigma^{2}\nabla_{x}\ln{\vec{p}}_{1-\tau }\left(x\right) \end{array}$$
(13)$$\begin{array}{cc} \vec{\mu }(x,t) & =-\overset{↼}{\mu }\left(x,1-t\right)+\sigma^{2}\nabla_{x}\ln{\overset{↼}{p}}_{1-t}\left(x\right). \end{array}$$

Keeping in mind the relationship between the forward time index *t* and reverse time index τ=1−t, ${\vec{p}}_{1-\tau }\left(x\right)$ is the marginal density of the state variable $X_{t}$. Likewise, the density ${\overset{↼}{p}}_{1-t}\left(x\right)$ corresponds to the marginal density of the backward state variable $Z_{\tau }$ evaluated for *x*. The superimposed harpoons indicate the flow of time with $\vec{\mu }$ being the drift of the forward process and $\overset{↼}{\mu }$ being the corresponding drift of the backward process. For the interested reader, we provide a derivation of the reverse-time drift resulting in the relationship above in Appendix A.

Figure 2 exemplifies visually how the reverse drift can be obtained from the respective forward drift and the score of the probability distribution over paths induced by the forward SDE. It is only possible in rare cases to compute this density analytically by solving the Fokker–Planck equation. Luckily, there is a numerical method *score matching* [14] that allows for a direct estimation of the gradient of log-densities in (Equation 12) from an ensemble of simulated data. This technique is well established in the field of machine learning.

This approach has been previously suggested in the literature, but deemed to be impractical [12] for a solution of the Schrödinger problem. The direct implementation of score matching to (13) in the IPF iterations would create considerable algorithmic problems. Every full iteration of the IPF algorithm would add a score term to the previous reverse drift, which in later iterations would itself be a sum of previous drifts and the score over the previous probability of paths that scales with each IPF iteration *i*.

If one represents both the score estimator and the current (e.g., the forward) drift by a nonlinear function approximation such as a neural network, the updated (backward) drift (Equation 12) becomes the sum of two neural networks, which is not easily represented as a single one. Hence, during the iterations, one would have to store the entire sequence of past drift functions in order to compute the present one. This would make the algorithm extremely complicated and slow, as we would have to keep in memory 2i score matching neural networks of the *i*’th IPF iteration. This would also mean that, for a single drift evaluation in the *i*’th IPF iteration, we would have to evaluate the 2i−1 and 2i neural networks for a single drift evaluation at the *i*’th IPF iteration. For this reason, the score matching approach has not been applied to the Schrödinger bridge problem.

Alternative approaches to computing the drift functions are based on the *Euler–Maruyama* (EM) [15] temporal discretisation of forward and backward SDEs. From its conditional Gaussian transition densities, one can obtain a likelihood function for the drift function evaluated at the discrete time points. Using samples obtained from a forward process, one can estimate the corresponding backward drift using a maximum likelihood or Bayesian approach. This method was applied to the generation of half-bridges by [13], where Gaussian processes were used as a prior distribution over functions. Reference [12] developed a different method that used the conditional Gaussian transition densities of the EM discretisation to approximate the score function. This expression can then be converted into an approximation (which becomes exact in the limit when time interval used for discretisation goes to zero) for the drift function. Both approaches could be viewed as methods for approximating the backward drift (Equation 9) using a small time interval *h* and by computing the conditional expectations within a regression framework. This approach needs strong regularisation, as denoted in [12], which required running averages of the entire function approximators to guard against fatal training divergences as the drifts were trained on local estimates dependent on the interval *h*, as in Equation (Equation 2).

In summary, previous approaches in [12,13] approximated the drifts with the expected infinitesimal change defined in Equation (Equation 2) in order to yield a locally tractable reverse drift, thus omitting the influence of the score necessary for the analytical reverse process. In this work, we propose to include a surrogate form of the score term in the reverse drift such that the respective reverse drifts are trained to approximate the complete reverse drift and not just its localised estimates. We will show in the following that the representation of the drifts (Equation 12) and (13) can be directly estimated in a straightforward way by a modification of the score matching approach.

### 2.4. Score Matching with a Reference Function

To simplify the notation, we denote by $\mu \left(x_{t},t\right):R^{D}\times \left[0,1\right]\to R^{D}$ one of the two drift functions and a corresponding marginal density $p_{t}\left(x\right)$ induced by the SDE with drift $\mu (x_{t},t)$ and a scaled Wiener process with constant diffusion σ. Following [22], we define the following cost functional of the smooth vectorial function $\phi \left(x,t\right):R^{D}\times \left[0,1\right]\to R^{D}$:(14)$$\begin{array}{cc} L[\phi ,\mu ] & =\int_{0}^{1}dt\int dxp_{t}\left(x\right)\left\{\phi {\left(x,t\right)}^{T}\phi \left(x,t\right)+2\mu {\left(x,t\right)}^{T}\phi \left(x_{t},t\right)+2\sigma^{2}\mathrm{Tr}\left[J_{\phi }\left(x,t\right)\right]\right\} \end{array}$$
where $\mathrm{Tr}[J_{f}\left(x\right)]$ is the trace of the Jacobian of a function $f\left(x\right):R^{D}\to R^{D}$. With respect to the Schrödinger bridge problem, μ(x,t) would be the drift of the reference process and ϕ(x,t) would represent its reverse-time process. We purposefully withheld the harpoons denoting the flow of time earlier, as each half-bridge in Equations (Equation 6) and (7) alternates its reference and reverse drift. This score matching with a reference function does not require access to the true score, but instead, uses a surrogate function that is constructed from readily available numerical quantities. By straightforward integration by parts, we can show (see Appendix B) that
(15)$$\begin{array}{c} \phi^{*}\left(x,t\right)≐\arg\min_{\phi }L\left[\phi ,\mu \right]\left(x,t\right)=-\mu \left(x,t\right)+\sigma^{2}\nabla_{x}\log p_{t}\left(x\right). \end{array}$$

Hence, a comparison with Equations (Equation 12) and (13) shows that the minimiser of the functional, for a given forward or backward drift, provides the corresponding reverse drift. For a practical computation of the cost function, the integrals over time and over the unknown density in (Equation 14) are approximated by numerically generating $N_{X}$ independent trajectories of the process sampled at $N_{t}$ regular time points $t_{j}$.

Hence, we approximated the cost function by its sample-based estimator:(16)$$\begin{array}{cc} \hat{L}\left[\phi ,\mu \right] & =\sum_{j=1}^{N_{t}}\sum_{i=1}^{N_{x}}\left\{\phi {\left(x_{t_{j}}^{\left(i\right)},t_{j}\right)}^{T}\phi \left(x_{t_{j}}^{\left(i\right)},t_{j}\right)+2\mu {\left(x_{t_{j}}^{\left(i\right)},t_{j}\right)}^{T}\phi \left(x_{t_{j}}^{\left(i\right)},t_{j}\right)+2\sigma^{2}\mathrm{Tr}\left[J_{\phi }\left(x_{t_{j}}^{\left(i\right)},t_{j}\right)\right]\right\} \end{array}$$

For a finite sample size, the empirical cost function must be regularised by controlling the complexity of the functions ϕ(·,·). In contrast to [22], we modelled $\phi \left(x,t\right):R^{D\times [0,1]}\to R^{D}$ by a *single nonlinear parametric* function, which is given by a multilayer neural network (rather working with time slices using a distinctive function of *x* for each). In such a way, we implicitly incorporated the smoothness of the drift in both space *x* and time *t*. The construction of the reverse drift and the use of function approximators is exemplified in Figure 3.

### 2.5. Numerical Considerations and Implementation Details

The trace of the Jacobian requires the evaluation of the derivative of a single output with respect to the single input in the same dimension *d*, independent of all other outputs and inputs. Since we evaluated a single drift jointly for all dimensions *D*, this makes the computation of the analytical Jacobian expensive for higher dimensions, as we have to perform *D* independent backward passes to compute each entry in the Jacobian matrix. The number of gradient computations required for the Jacobian in a vector-valued function thus scales quadratically with the number of dimensions.

For data with few dimensions, computing the diagonal terms of the Jacobian can be performed via batched the backpropagation of one-hot output gradients. This approach falters computationally and memorywise when we consider data in higher dimensions. For higher-dimensional data, we opted for the trace estimation trick of Hutchinson with samples from an i.i.d. Rademacher distribution in $R^{D}$, namely
(17)$$\begin{array}{c} \mathrm{Tr}\left[J_{\phi }\left(x_{t},t\right)\right]=E_{z∼p(z)}\left[z^{T}\nabla_{x}\left[\phi {\left(x_{t},t\right)}^{T}z\right]\right]. \end{array}$$

An elaboration on the trace estimation trick can be found in Section C.1. The main idea of the stochastic approximation of the trace is that we are only interested in the diagonal elements of the Jacobian. The Hutchinson trace estimation trick proceeds by computing the derivative with respect to the input of the inner product of the prediction $\phi (x_{t},t)$ and a random variable *z*. The same random variable is then applied a second time in an inner product to obtain an approximation of the scalar quantity of the trace of the Jacobian. The advantage of the trace estimation trick is that only a single additional derivative evaluation on the inner product $\phi {\left(x_{t},t\right)}^{T}z$ is required, which scales linearly with the number of samples of *z*.

Taking the gradients of the loss with respect to the parameters requires a second derivative such that a function approximator trained on the reverse drift has to be twice-differentiable. Enforcing this property in neural networks requires us to use at least twice-differentiable evaluations of the prediction with respect to the spatial input $x_{t}$, which necessitates twice-differentiable activation functions such as the hyperbolic tangent or Gelu activation functions [23].

If the function approximator is only once differentiable as with the use of rectified linear unit activation functions [24], we can employ Stein’s lemma to estimate the trace of the Jacobian. For this estimator, following [25,26], we define an isotropic Gaussian perturbation distribution $z∼N(x_{t},\sigma_{z}^{2}I)$ with $x_{t},z\in R^{D}$ around each data point $x_{t}\in R^{D}$ and average the gradients in the *z*-neighbourhood of the data point:(18)$$\begin{array}{c} \mathrm{Tr}\left[J_{\phi }\left(x_{t},t\right)\right]=\lim_{\sigma_{z}↓0}E_{p(z)}\left[\phi {\left(x_{t}+z,t\right)}^{T}\frac{z}{\sigma_{z}^{2}}\right] \end{array}$$
The derivation of Stein’s lemma can be followed up in Section C.2, and its application to the trace estimation therefrom is detailed in Section C.3. For a practical implementation, we approximated the Stein estimator using a sufficiently small $\sigma_{z}$ by drawing only a single random vector $z_{j}^{\left(i\right)}$ for each trajectory *i* and each time point *j*. The perturbed ϕ function values can be computed along side the unperturbed values in a single forward pass through the neural network.

## 3. Results

We evaluated our proposed method on artificially generated datasets with varying dimensions and with and without dependencies between the dimensions at the two target marginals. The first set of experiments were performed on the construction of the Schrödinger bridge between Gaussian mixture models with which we could explore the behaviour with a changing set of dimensions. The second collection of experiments focused on manifold learning in which implicit distributions were learned to be generated from a standard normal distribution. Finally, we employed our proposed framework on the generation of intermediary distributions of embryoid single-cell RNA as a real-world application.

### 3.1. Experimental Setup

For all our experiments, we used a deep neural network, taking both the spatial input $x_{t}$ and the time *t* as distinct inputs. We fixed the size of the fully connected layers in the hidden layers of the deep neural networks to an integer multiple of the spatial dimension *D*. As a rule of thumb, we used 10D neurons in the hidden layers and scaled the depth of the deep neural network with max(2,D/5).

We used LayerNorm [27] before the spatial features of the hidden layers so as to not destroy the time embeddings. LayerNorm normalises the representation of each sample to a standard normal distribution and has empirically been shown to numerically aid the gradient computation. Thus, we used blocks of the shape $x_{i+1}$ = $x_{i}$ + Tanh(Linear( LayerNorm($x_{i}$), Embedding(t))). We used the Adam optimiser [28] and cosine annealing [29], training each half-bridge for 1000 steps and annealing the learning rate from $10^{-3}$ to $10^{-5}$. We drew $N_{x}=128$ trajectories per bridge and stored them in a buffer of 512 sample trajectories, discarding old trajectories as needed to maintain the fixed buffer size. We constructed the Schrödinger bridge by running 10 IPF iterations.

The simulations of the SDE were performed using the *Euler–Maruyama* [15] approximation with a step size dt with $N_{t}$, and we found $N_{t}=100$ and dt=0.01 to be robust values working well for our experiments. Our choice of the diffusion σ was motivated by the idea that the samples of the half-bridge process at the first IPF iteration should sufficiently cover the marginal distribution, increasing the possibility that this distribution is “hit” with at least some sampled trajectories. As the diffusion parameter is yet another hyperparameter, we chose the diffusion according to $\sigma =MM1\;/\;N_{t}\cdot dt$.

The drifts of both processes, forward and backward, received as input the spatial information *x* and the time *t*. We normalised the time index $t\in [0,N_{t}\cdot dt]$ to t∈{0,1} as the time index remained fixed over the course of the entire training of the bridge.

The question remains how the reference process $Q_{0}^{*}$ should be chosen for our applications. The authors in [12] used an Ornstein–Uhlenbeck (OU) process [30] for $Q_{0}^{*}$. Its marginal distributions can be computed analytically and do not require solving an SDE numerically, which saves time and computational resources during the very first half-bridge. This choice of a Gaussian reference process could be also motivated from the fact that, for larger times *t*, the marginal of the process converges to a stationary Gaussian density that could approximately match Gaussian targets used for a denoising-style data generating application of Schrödinger bridges.

In our implementations, we did not want to make any specific assumptions on the end marginals. Hence, a choice of a zero initial drift μ(x,t)≡0 (reducing $Q_{0}^{*}$ to a simple Wiener process) seemed more natural. However, practical considerations suggested a slightly different approach, in which $Q_{0}^{*}$ corresponds to a a drift function represented by a neural network that has (untrained) small random weights, which serve as useful initial conditions for the subsequent training [31,32]. We observed experimentally that for the first half-bridge $Q_{0}^{*}$, the Wiener process dominates the characteristics of the sampled trajectories.

For our applications, the marginal densities at the initial and end times ${\vec{p}}_{0}$ and ${\vec{p}}_{1}$, which are generated by the bridge models, as well as the desired targets $\pi_{0}$ and $\pi_{1}$ are represented by random samples, rather than by analytical expressions. In order to evaluate the quality of the converged bridge, we computed the Wasserstein-1 distances $W_{1}\left({\vec{p}}_{1},\pi_{1}\right)$ and $W_{1}\left(\pi_{0},{\overset{↼}{p}}_{0}\right)$. The Wasserstein-1 distance [33] between two probability measures μ and ν is defined as
(19)$$\begin{array}{c} W_{1}\left(\mu ,\nu \right)=\underset{\gamma \in D(\mu ,\nu )}{\inf}E\left[||x-y||\right] \end{array}$$
where D(μ,ν) is the set of all couplings of μ and ν. These can be straightforwardly evaluated on empirical distributions, which are given by samples. For the underlying optimal transport problem and its efficient solution via linear programming in its dual representation, see, e.g., [17]. The Wasserstein-1 distance is also known as the Earth Movers’ Distance (EMD) in computer science.

### 3.2. Multimodal Parametric Distributions

We modelled the marginal distribution $\pi_{0}\left(x\right)$ as a Gaussian distribution with a diagonal covariance matrix. The opposite marginal distribution $\pi_{1}\left(x\right)$ was a Gaussian mixture model with two modes with a uniform prior over the GMM component centres. The mean values of all Gaussian distributions, uni-modal in $\pi_{0}\left(x\right)$, as well as bi-modal in $\pi_{1}\left(x\right)$, were sampled uniformly from U(−2.5,2.5), and a standard deviation of 1.0 was used throughout. The visualisation of the inferred Schrödinger bridge highlights the learning of the time-dependent drift and the ability to model bifurcations in the case of bi-modal GMMs, as seen in Figure 4.

The dataset was created as the more tractable experiment in comparison to subsequent datasets. The use of GMM’s allows for the analytical evaluation of the probability of the generated data at the marginal distributions. Furthermore, the modes of the GMM could be handcrafted, which turned out to be important to validate numerous design choices of the drift approximators. The authors deemed it equally important to visually verify the generated trajectories, ensuring that the drift approximators were able to model, for example, bifurcations and how they dealt with changing hyperparameters such as increased diffusion.

Estimating the score in regions with little probability mass is a difficult problem, as an insufficient amount of samples may be drawn from that region to accurately model the score. This was explored in detail in [1,34] and was one of the inspirations to the perturbation protocol in diffusion models. Diffusion models have the advantage of computing an analytical score at any point in space and time through their analytical perturbation kernels defined in the forward process.

The Schrödinger bridge problem offers none of these luxuries, as no reference process is available that yields analytical scores. Thus, we require sufficient data even in low-probability regions to enable us to estimate the necessary score. We found in our experiments that the hyperparameter with the single largest influence was the number of trajectories that were sampled from the path measures. We can thus see in Figure 5 that increasing the number of trajectories decreases the Wasserstein-1 distance with respect to the marginal distributions $\pi_{0}\left(x\right)$ and $\pi_{1}\left(x\right)$ as the score estimation becomes more precise, as even low-probability regions of the stochastic processes, on which the score is estimated, are sampled adequately.

We chose the multi-modal dataset as a test dataset to compare Hutchinson’s trace estimator with the trace estimation via Stein’s lemma. The result can be seen in Figure 6, in which we evaluated the Wasserstein-1 distance over a fixed set of dimensions. We observed that the Wasserstein-1 distance increases linearly with the number of dimensions for a fixed number of samples, and it was interesting to observe that the Schrödinger bridges with different trace estimators behaved very similar in terms of performance. We compared Stein’s trace estimator as detailed above and in Section C.3 with varying perturbation scales with Hutchinson’s trace estimation and the ground truth Wasserstein distance between the marginal distributions $\pi_{0}\left(x\right)$ and $\pi_{1}\left(x\right)$. In theory, Hutchinson’s trace estimation can also be applied with a normal distribution sampling the random projection vectors, yet Rademacher’s distribution has the lowest estimator variance. Stein’s trace estimation can only be performed with the normal distribution, as it relies on integration by parts and the special derivative of the normal distribution. This led us to hypothesise that the inherently higher variance of the normal distribution in Stein’s trace estimator leads to worse performance. However, we aim at examining this in future work.

### 3.3. Manifold Datasets

For the second dataset, we trained the forward and backward drifts on the generated manifold data via the Sklearn machine learning package [35]. The manifolds used were “make_swiss_roll”, “make_s_curve”, and “make_moons” from the sklearn.dataset code base, which were concatenated to create a higher-dimensional manifold. This increased the complexity of the manifold, setting it apart form earlier manifold modelling approaches, as in [12].

We trained the stochastic processes to predict multiple manifolds at once by modelling them jointly with a single fully connected neural network. For $\pi_{0}\left(x\right)$, we chose a standard normal multivariate distribution $\pi_{0}\left(x\right)=N\left(0,I\right)$, while $\pi_{1}\left(x\right)$ was the implicit distribution generated by samples on the manifold. Whereas the previous Gaussian mixture models were statistically independent in each dimension, the manifolds explicitly modelled the statistical correlation between different dimensions. A visualisation of the Schrödinger bridge between the two distributions can be seen in Figure 7.

The use of a tractable probability distribution as one marginal distribution was inspired by the purely generative task of diffusion models. The dataset is commonly used as a visual benchmark of new generative models and allows evaluating the drift approximators’ ability to model nonlinear manifolds. A lack of this dataset is the absence of a tractable data likelihood under the marginal distribution, as the manifold generation function is modelled as an implicit distribution.

### 3.4. Embryoid Dataset

Single-cell RNA sequencing analyses the RNA of individual cells, destroying it unfortunately and making it inaccessible for further analysis. In a population of cells, we can remove individual cells and analyse their RNA. As each cell is eliminated from the population, we have to turn to a probabilistic method to simulate the development of the RNA at the *population* level.

Therefore, it is of interest to develop a methodology that can simulate the full trajectories of RNA sequences over time, which would allow for predicting the outcomes of such measurements on a single cell without actually having to perform them. As suggested in [36], an interesting solution to this problem would be the construction of a stochastic generative model for the possible measurements with the marginal distributions (for a population of cells) of the actual measurements at the initial and end time as boundary conditions. A visualisation of the application of the Schrödinger bridge to the RNA measurement task is provided in Figure 8. If we represent this generative model by a diffusion process, we naturally end up with the idea of applying Schrödinger bridges to this problem.

We applied this approach to the embryoid dataset [37] for which single-cell RNA measurements were taken at five different times t∈{0,1,2,3,4}. The ensemble of measurements at each time index *t* consists of a varying number (2380, 4162, 3277, 3664, 3331) of RNA measurement samples. We considered the problem of constructing a Schrödinger bridge using only the first day and the very last day measurements. We then tried to infer the hold-out intermediate marginal distributions of the RNA measurements with the help of the constructed bridge. The samples drawn from the bridge were thus evaluated at the intermediate marginal distributions at t∈{1,2,3} using the Wasserstein-1 distance. In addition, we also evaluated the Schrödinger bridge at the two end-points by computing $W_{1}\left({\overset{↼}{p}}_{0},\pi_{0}\right)$ and $W_{1}\left({\vec{p}}_{1},\pi_{1}\right)$.

We compared our method with two other generative model approaches: *TrajectoryNet* [37] implements constrained normalising flows using neural networks. The paper was also the first to propose this benchmark. IPML [13] presents an alternative method to constructing a Schrödinger bridge in which the required drift functions are estimated from the trajectories using Gaussian process regression without computing score functions, as remarked upon in Section 2.3. Finally, we have also included a simpler method, which is based on Optimal Transport (OT). It computes a linear transport map between samples of the initial and end distributions. However, the comparison of this method with the others is slightly unfair. While the OT approach allows for making predictions of the marginals at intermediate time steps, it is not formulated as a generative model and, thus, could not be used to make predictions for measurements on single cells.

The RNA sequencing data were generated from FACS-sorted embryoid bodies via surface marker indication. The dataset was preprocessed with PHATE [36] to a dimensionality of D=5, as performed in the reference methods. An important feature of the PHATE preprocessing is the reversibility of the dimensionality reduction. The preprocessing pipeline can be accessed and replicated with the public PHATE code base provided by the authors of [36]. Thus, it is possible to apply trajectory reconstructing algorithms in the low-dimensional representation of the genetic data and project the resulting trajectory back into the high-dimensional space. This is in contrast to commonly used dimensionality reduction algorithms, which do not offer these advantages. We refer the reader to [36] for an in-depth treatment of the algorithm. The experimental setup was kept identical for *IPFML* and TrajectoryNet.

Table 1 compares the performance of our Schrödinger bridge method with the other methods. Our method outperformed both Trajectorynet and the alternative Schrödinger bridge method IPML and was also better than the OT approach in two out of three instances. This corroborates the result from [13] that OT’s linear transport map is not a well-fitting intermediate distribution for t=4. Especially, the Wasserstein distance at the marginal distributions of the backward process at t=0 and the forward process t=4 were well modelled.

## 4. Discussion and Conclusions

We presented a method for solving the Schrödinger bridge problem based on the iterative proportional fitting algorithm. In contrast to the simpler diffusion models for data generation, Schrödinger bridges can provide interpolations between *two* arbitrary data distributions. Unfortunately, this advantage comes with the drawback that the score functions that can be used to compute drift functions for the time-reversed process are not analytically available. The novel aspect of our approach is the formulation of reverse-time drift functions as solutions of a minimisation problem (an extension of the score matching method) involving expectations over the forward process. This allows for an efficient training of neural networks representing the drift functions on simulated trajectories of the corresponding stochastic differential equations.

To evaluate our approach, we conducted experiments on two synthetic datasets, which allowed us to analyse its performance within a controlled environment. Finally, we applied our method to a single-cell mRNA dataset in which a Schrödinger bridge was constructed between intermediate measurements, where the initial and end distributions were both non-Gaussian. On this dataset, we outperformed similar methodologies, also beating a linear transportation map in two out of three instances.

Our variational approach of estimating drift functions in the IPF algorithm for solving Schrödinger bridges could be extended in various ways:The variational formulation of our drift estimators is independent of the temporal discretisation used for creating sample trajectories. It only depends on the *marginal distributions of the state variables*. This fact opens up alternative possibilities for generating appropriate samples by forward and backward simulations, which could allow for larger stepsizes. One might, e.g., consider *weak* approximation schemes [38] for numerically simulating SDEs. A different alternative is the application of *deterministic*, particle-based simulations [22], where the reduced variance of estimators might allow keeping the number of trajectories small. Finally, *exact sampling* methods (see, e.g., [39]), which entirely avoid temporal discretisation, would be interesting candidates.Reliably estimating the score-based drift functions in regions with small marginal probability densities remains a challenging problem, especially for higher dimensions. This is evident in our synthetic experiments, as the number of trajectory samples remained the most-important hyperparameter to ensure the convergence of the Schrödinger bridge (see also [34] for similar observations). It would be interesting to see if prior knowledge expressed by exact analytical results for asymptotic scaling of densities could be implemented in the function approximators to improve on that problem.It is relatively straightforward to adapt the variational approach (see e.g., [22]), together with a corresponding change in the relations (Equation 12) and (13) to solve Schrödinger bridge problems for more general types of stochastic differential equations. Interesting cases could include SDEs with (fixed) state and time-dependent diffusion matrices, as well as processes based on *Langevin dynamics* (i.e., systems of second-order SDEs), well-known for modelling physical systems that are also used for Hamilton Monte Carlo simulations.It would be interesting to investigate possible simplifications of our method for the special case in which the drift of the reference process is the gradient of a potential function. The relations (Equation 12) and (13) show that all half-bridges in the IPF algorithm will inherit this property. One could then modify the variational formulation and learn directly the potential, rather than the *D* components of its gradient individually. This built-in symmetry might increase the accuracy of estimation, but the need for second derivatives in the cost function could lead to an increase of the numerical complexity of training a neural network.

## Figures and Tables

**Figure 1 entropy-25-00316-f001:**
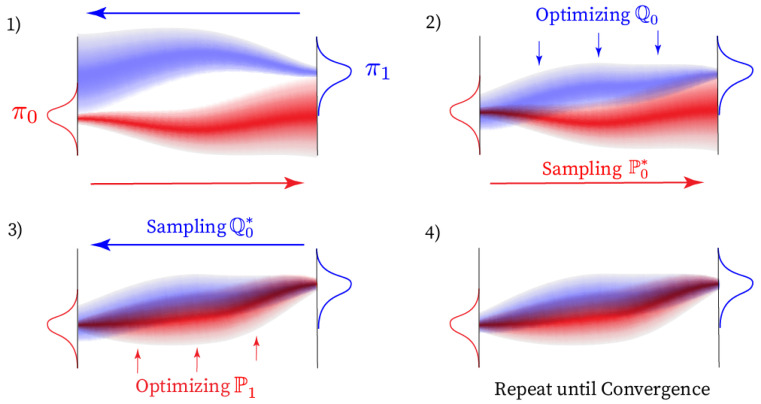
Visualization of the convergence of a one dimensional Schrödinger bridge problem via an Iterative Proportional Fitting style optimization. Subplot 1) shows the initial forward $P_{0}^{*}$ and backward $Q_{0}^{*}$ in red and blue. In the first half-bridge in subplot 2), $P_{0}^{*}$ is held fixed and $Q_{0}^{*}$ is obtained by optimizing equation (7). Consequently, corresponding to equation (Equation 6), $P_{1}$ is fitted on a constant $Q_{0}^{*}$ in subplot 3). This procedure is repeated until both $Q_{i}^{*}$ and $P_{i}^{*}$ converge according to some predetermined criterion as indicated by subplot 4).

**Figure 2 entropy-25-00316-f002:**
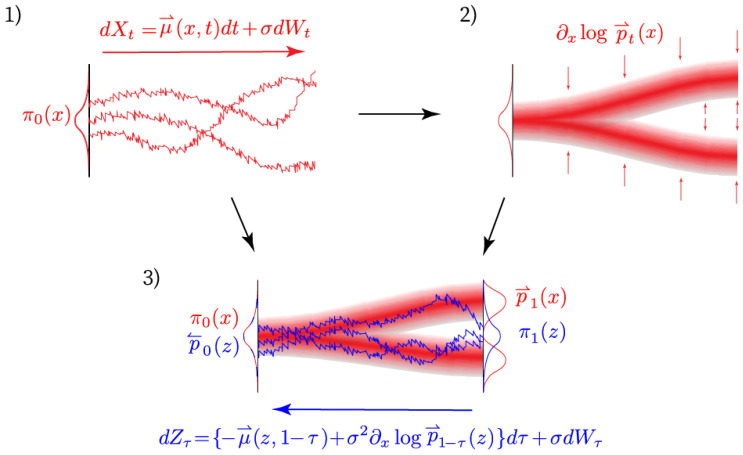
Visualization of the construction of the reverse process. The subplot 1) exemplifies three trajectories generated by solving the forward stochastic differential equation $dx_{t}$. The path measure P induces a probability distribution ${\vec{p}}_{t}\left(x\right)$ from which the score is estimated in subplot 2). Finally subplot 3) shows three possible trajectories of the reverse process starting from $\pi_{1}$ and finishing in $\pi_{0}$. A simpler visualization of the score can be found in subplot A), which serves as a figurative illustration of the behaviour of the score $-\partial_{x}\log p\left(x\right)$ on a simple one dimensional distribution.

**Figure 3 entropy-25-00316-f003:**
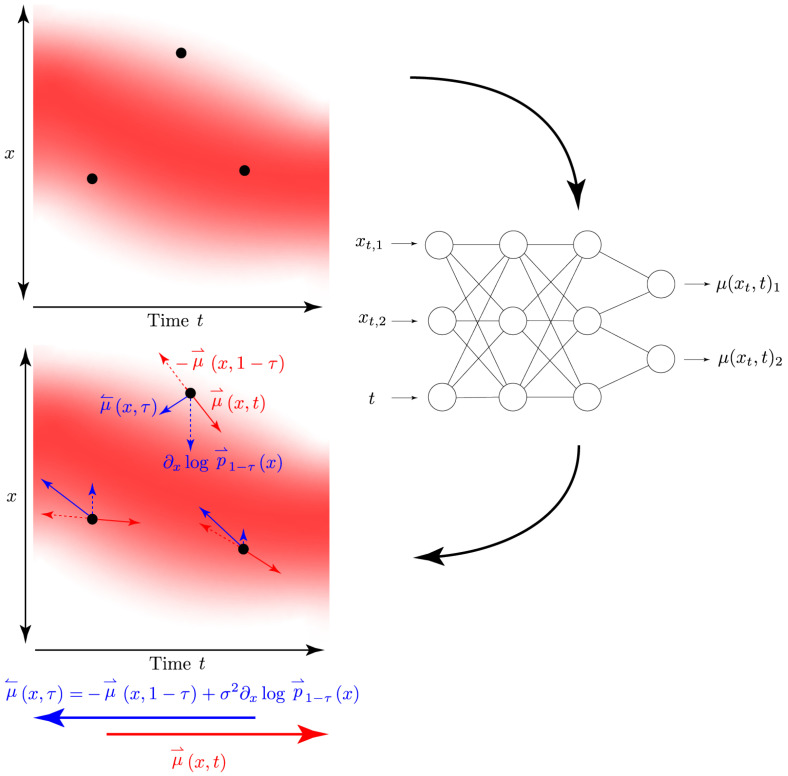
Visualisation of the construction of the backward stochastic process parameterised by the backward SDE with drift $\overset{↼}{\mu }(x,\tau )$. The red gradient represents the marginal distribution ${\vec{p}}_{t}\left(x\right)$ induced by the forward SDE with drift $\vec{\mu }(x,t)$ and marked in the colour red. We employed neural networks to learn both the forward and backward drift, as it allows for a single function approximator per process for the entire input domain as neural networks are inherently able to model vector-valued data.

**Figure 4 entropy-25-00316-f004:**
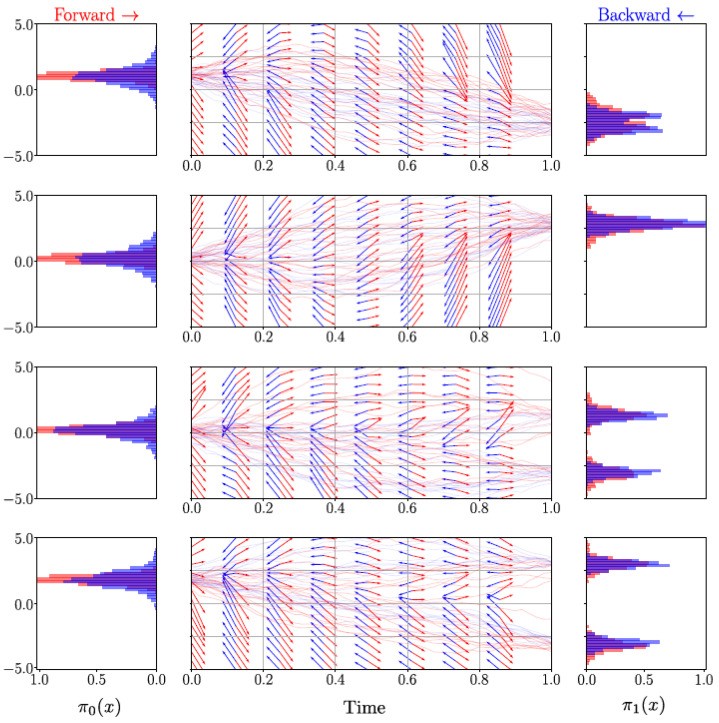
Visualisation of a solved Schrödinger bridge in $R^{4}$ in which each dimension is plotted independently. The sampled trajectories and drift of the forward process and its initial condition $\pi_{0}$ are shown in red, whereas the backward process is shown in blue.

**Figure 5 entropy-25-00316-f005:**
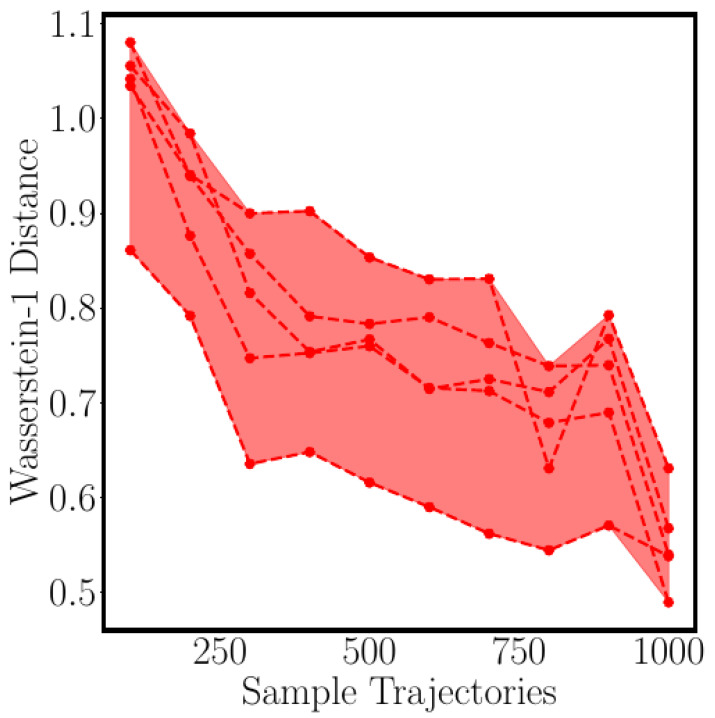
A comparison of the Wasserstein-1 distance on the number of sample trajectories sampled per IPF iteration for the multi-modal distribution problem in Section 3.2. The Wasserstein-1 distances $W_{1}\left({\vec{p}}_{1},\pi_{1}\right)$ and $W_{1}\left(\pi_{0},{\overset{↼}{p}}_{0}\right)$ were averaged and show an overall decrease relative to the number of trajectories sampled from the stochastic processes.

**Figure 6 entropy-25-00316-f006:**
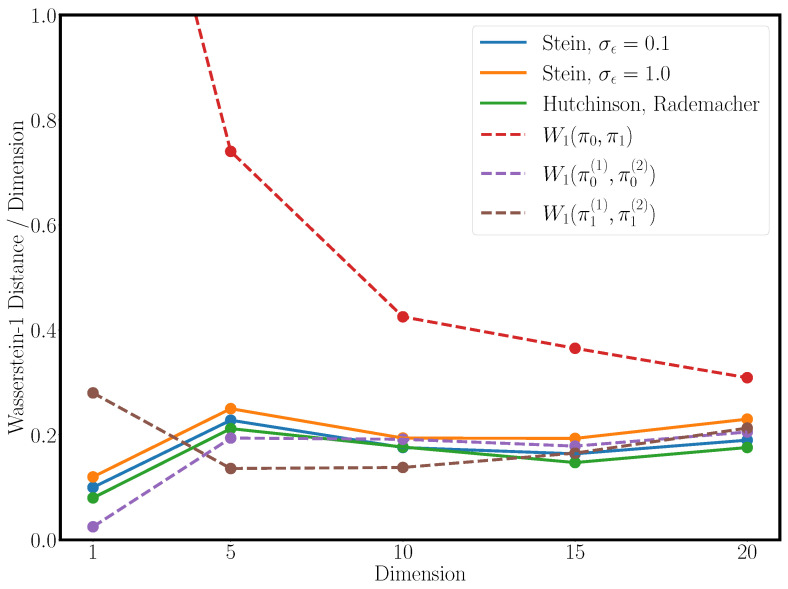
A comparison of the gradient estimators on an increasing number of dimensions with the average Wasserstein-1 distance per dimension between the true marginal and the predicted marginal distributions of the forward and backward process. As baselines, the Wasserstein-1 distances between the respective marginals is shown. One can see that the gradient estimators performed as well as the Wasserstein-1 distances between samples drawn from the marginals denoted as $W_{1}\left(\pi_{0}\left(x\right),\pi_{0}\left(x\right)\right)$ and $W_{1}\left(\pi_{1}\left(x\right),\pi_{1}\left(x\right)\right)$. The Hutchinson trace estimator performed best while requiring a second derivation. Interestingly, smaller sampling variances for the Gaussian distribution in the Stein gradient estimator yielded better overall performance on matching the marginal distributions.

**Figure 7 entropy-25-00316-f007:**
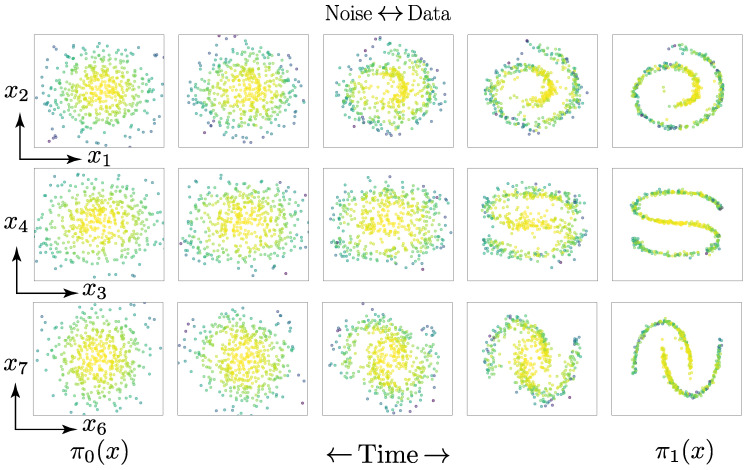
Visualisation of a constructed Schrödinger bridge for the manifold $R^{6}$, which lies in $R^{10}$. The left-most column represents samples from the marginal distribution $\pi_{0}\left(x\right)$, which are transformed into samples on the manifolds in the right-most column. The colour coding of each particle is according to its proximity to the mean of the prior distribution $\pi_{0}\left(x\right)$ such that we can distinguish which particles from $\pi_{0}\left(x\right)$ correspond to the particles on the manifold $p(x_{1},1)$.

**Figure 8 entropy-25-00316-f008:**
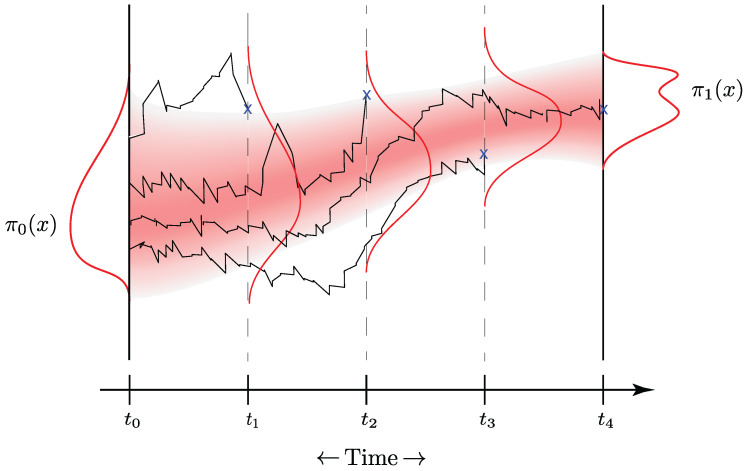
A visualisation of how the Schrödinger bridge is applied to the RNA measurements. The RNA of different cells is measured at the discrete time steps $t_{1}$, $t_{2}$, and $t_{3}$ and eliminated from the population marked by the blue crosses and the end of each of the four trajectories. Given these snapshots of the RNA distribution at discrete time steps, the task is now to construct a Schrödinger bridge between the discrete time steps as indicated by the stochastic process with a fading colour. The generative model can then be queried at any time step between $t_{0}$ and $t_{4}$.

**Table 1 entropy-25-00316-t001:** Comparison of comparable methodologies on the embryoid dataset with the Wasserstein-1 distance The “Path” column denotes the average EMD of the intermediate time steps T∈{2,3,4}, whereas the “Full” column averages all time steps. The optimal transport linear transport map is only defined for the intermediate time steps, as it requires the two marginal distributions at T∈{1,5}.

Method ↓	t = 1	t = 2	t = 3	t = 4	t = 5	Path	Full
TrajectoryNet	0.62	1.15	1.49	1.26	0.99	1.30	1.18
IPML EQ	0.38	1.19	1.44	1.04	0.48	1.22	1.02
IPML EXP	0.34	**1.13**	1.35	1.01	0.49	1.16	0.97
OT	N/A	1.13	1.10	1.11	N/A	1.16	N/A
Reverse-SDE	**0.23**	1.16	**1.00**	**0.64**	**0.28**	**0.93**	**0.66**

## Data Availability

The RNA data can be found in this github repository.

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
