# Peer review of "A Score-Based Approach for Training Schrödinger Bridges for Data Modelling"

_entropy, 2023, doi:10.3390/e25020316_

Round 1
Reviewer 1 Report
English language and writing style is outstanding.
Line 357 should be “MO and CO have been partially funded”
Lines A43-A46. Why are parts in red?
Author Response
We thank the reviewer for their time they took to read and improve our manuscript.
- The typo in line 357 has been corrected. Thank you for pointing this out.
- The red highlight has removed and replaced with underbraces and enumeration to highlight what terms are combined into the subsequent terms.
Reviewer 2 Report
This paper introduce a modified score function based method for computing a drift function which will be used for Schroedinger bridge's computational training. This function can be efficiently implemented by a feed–forward neural network. This approach is applied to artificial data sets with increasing complexity . Its performance on genetic data has been evaluated in this paper.
In general, this paper is interesting and well organized. But some improvements and revisions have to be made.
1. The background and research status of Schroedinger bridge should be explained in detail in the introduction.
2. The main contributions need to be further refined.
3. Page 2. The Schroedinger bridge problem is a control problem that needs to be explained more clearly.
4. Page 3. The introduction of the previous methods used to solve the Schrodinger bridge problem needs to be more concise.
5. Page 7. "higher" need to be changed to "Higher".
6. Page 8. The source of the manually generated dataset needs to be explained.
7. The simulation examples in this paper are very great, only need to modify the simulation diagram more clearly.
8. References should be updated in time and language should be checked again.
9. You should strengthen the elaboration of future works in the conclusion section.
I would reconsider the revised manuscript after the authors answering the above questions.
Author Response
We thank the reviewer for their time reading and making suggestions for improving our manuscript.
We will now highlight the improvements based upon the reviewers remarks.
- We added a new paragraph (3. paragraph in introduction) which refers the reader to the research status of the Schrödinger Bridge Problem which provide references to the most recent reviews and theoretical background.
- We refined our contribution in the last paragraph of the introduction and in the last paragraph of section 2.3.
- We have provided appropriate references to the control problem formulation in the third paragraph of the introduction by referencing "A survey of the Schrödinger problem and some of its connections with optimal transport" by Leonard and " On the relation between optimal transport and Schrödinger bridges: A stochastic control viewpoint" by Chen.
- Page 3 serves as the mathematical foundation of the problem and aims at rigorously presenting the problem we want to solve. Remarks to previous methods have been removed as much as possible.
- Due to the rewriting the typo has been resolved.
- We added explicit references to the functions that were used in scikit learn to generate the data manifolds (’make_swiss_roll’, ’make_s_curve’ and ’make_moons’)
- We have added more labels to all relevant figures. All marginal distributions (\pi_0/1(x)) are now properly labeled and a 'Time' label has been added. The manifold data has now axis labels to better highlight the manifold relevant data.
- Has been done. Thank you for the remark.
- Future work has been rewritten to highlight specific approaches to improving the methodology in future work. Thank you for pointing out how to strengthen this part of the manuscript.
Reviewer 3 Report
Finding a mapping connecting two different probability densities is one of the fundamental problems of modern ML. The optimal transport problem gives a general framework and the performance of GAN relies on the excellent performance of neural networks for this problem.
The Schroedinger Bridges, which stands on an established mathematical background, is another formulation of the problem. One of its characteristics is that it enables us to compute the intermediate states between two densities naturally. It is a new direction in ML. We find major approaches are emerging to solve this problem as in [8,13].
Compared to them, the main proposal of the submitted paper is to use a neural network to estimate the drift function. At this stage, we should welcome different proposals for this new direction in ML. However, the present draft is somewhat imbalanced. For example, many equation expansions in appendixes are trivial and lengthy, while discussions and explanations of the experiments are insufficient. I do not think the authors need additional numerical experiments but many parts of the paper should be rewritten before another round of review. I list some comments below.
Major comments:
-
The submitted paper shares the basic mathematical formulation of the problem with [8] and [13]. In the latter half of page 4, the authors describe the difference between the proposal and [8], but I do not understand the point clearly. It seems to me that the authors can make their points clearer by reorganizing sections 2.1 - 2.3. For example,
-
First, show the common (with [8] and [13], for example) basic mathematical formulation of the problem compactly.
-
Show what is the essential new proposal of the paper by comparing it with other proposals.
-
The descriptions and discussion of the experiments are somehow insufficient. It is not possible to reproduce the results. The authors should provide sufficient information for the setup of them and also discuss the “Pros and Cons” of the proposed method.
-
For the experiment in 3.3, I do not understand how you varied the dimensionality. If you used some setup in sklearn, it is better to show how you generated the data, or share your code.
-
For the RNA data, preprocessing is very important. It seems the authors followed [33] but more details should be provided. How they preprocessed RNA data to D=5 data? What is the physical meaning of “x”? I do not understand Figure 8 (what is the vertical axis?).
-
Many equation expansions in appendixes are elementary or almost trivial. Remove them and make appendixes compact. For example,
-
(A14) - (A65): Some are redundant and many of them are elementary.
-
(A66) - (A73): Most are trivial.
-
(A76) - (A79): You do not need these.
-
Appendix C.1: You do not need a page for this. Most equation expansions are redundant.
-
(A98) - (A100): This part is redundant.
-
(A101) - (A107): Remove some equations.
Minor comments
-
“Shroedinger” (title) or “Schrödinger” (Introduction, second paragraph)?
-
British/American spellings are mixed. I do not know the preference for “Entropy,” but be consistent at least.
-
Some definitions are redundant. The authors may think it is better to stay with them for better understanding, but you should denote how they are related to each other. For example, from Eqs. (12) and (13), \overrightarrow{p}_t(x) and \overleftarrow{p}_t(x) should have a direct relationship.
Author Response
We thank the reviewer for taking the time to read our manuscript and propose improvements to it.
We now want to address the reviewers remarks.
Major Comments:
1.1. In accordance to references [8] and [13] and upon checking also in the same mathematical form, the problem has been presented in half a page in a new section 2.2.
1.2. At the end of section 2.3 'Iterative Proportional Fitting in Schrödinger Bridges' we added a summary paragraph that highlights the inclusion of the score term in estimating the reverse drift whereas previous approaches omitted the score term for a more tractable regression target for the drifts.
2.1. Thanks to the very attentive reviewer we moved the score estimator comparison chart to its correct position in the Multi-Modal Parametric Distribution section which can be scaled arbitrarily. Thank you for pointing out this flaw. In accordance with other reviewers, we added the specific functions that were used in the Sklearn package to generate the manifold data (’make_swiss_roll’, ’make_s_curve’ and ’make_moons’ from the sklearn.dataset code base). Finally, we added a short discussion on the use of the generated data sets, pertaining to their advantages and disadvantages.
2.2. We rewrote the the paragraph on the preprocessing of the genetic data and linked the relevant publication as well as the code base of the preprocessing algorithm. We refrain from explaining the entire PHATE algorithm as it is beyond the scope of this manuscript and instead linke the ready to use package of the original authors of PHATE. We also highlight the advantage of using PHATE as a 'non-destructive' dimensionality reducer in comparison to t-SNE and PCA which do not allow the projection of low dimensional representations back into the original higher dimensional data space.
Figure 8. has been updated in accordance with other reviewers request and now the axes are clearly labeled in accordance to the variables used in the main text. The explanation of the blue crosses has also been clarified.
3.1. (A14) - (A65): While we do see the verbosity of the derivation, we believe that the appendix is a suitable place for the derivation of the reverse-time SDE to be written down. We would like keep the derivations, as the whole derivation of the reverse time drift has not been published in the machine learning community in its full length. We think that the full derivation is a useful help for undergraduate and graduate students from the machine learning community first encountering the reverse-time stochastic differential equations.
(A66) - (A73): We had multiple internal feedback sessions in which we had to repeatedly explain the derivation of the modified score matching. The majority of the feedback audience from the machine learning community and another reviewer were grateful for the detailed, yet verbose, derivation of the criterion. We thank the reviewer for going through the derivations attentively and seeing the their straightforward nature, we would still like to keep those derivations as people unfamiliar with the subject were grateful for the detailed explanation.
(A76) - (A79): We removed the equations as their are self-explanatory. Thank you for pointing this out.
Appendix C.1: We have indeed reduced the length of the derivation, sticking to the main points. Thank you for the feedback.
(A98) - (A100) and (A101) - (A107): They have been shortened.
Minor Comments:
Schrödinger has been standardized to the German Umlaut.
The text has been standardized to American spelling as far as we can see as non-native speakers.
We have added an explanation below Eqs. (12) and (13) for the visual help of distinguishing the forward from the backward process.
Sincerely, the authors
Round 2
Reviewer 2 Report
The revised work is ok, the reviewer does not have new comment.
Reviewer 3 Report
I still see too many equations in Appendix A. They are just undergraduate homework level. I leave the final decision to the editor.